# Sleep Spindle-Related EEG Connectivity in Children with Attention-Deficit/Hyperactivity Disorder: An Exploratory Study

**DOI:** 10.3390/e25091244

**Published:** 2023-08-22

**Authors:** Caterina Zanus, Aleksandar Miladinović, Federica De Dea, Aldo Skabar, Matteo Stecca, Miloš Ajčević, Agostino Accardo, Marco Carrozzi

**Affiliations:** 1Institute for Maternal and Child Health—IRCCS “Burlo Garofolo”, 34137 Trieste, Italy; caterina.zanus@burlo.trieste.it (C.Z.); marco.carrozzi@burlo.trieste.it (M.C.); 2Department of Engineering and Architecture, University of Trieste, 34127 Trieste, Italymajcevic@units.it (M.A.); accardo@units.it (A.A.); 3Department of Life Science, University of Trieste, 34127 Trieste, Italy

**Keywords:** sleep spindle, connectivity, graph analysis, EEG, ADHD

## Abstract

Attention-deficit/hyperactivity disorder (ADHD) is a neurobehavioral disorder with known brain abnormalities but no biomarkers to support clinical diagnosis. Recently, EEG analysis methods such as functional connectivity have rekindled interest in using EEG for ADHD diagnosis. Most studies have focused on resting-state EEG, while connectivity during sleep and spindle activity has been underexplored. Here we present the results of a preliminary study exploring spindle-related connectivity as a possible biomarker for ADHD. We compared sensor-space connectivity parameters in eight children with ADHD and nine age/sex-matched healthy controls during sleep, before, during, and after spindle activity in various frequency bands. All connectivity parameters were significantly different between the two groups in the delta and gamma bands, and Principal Component Analysis (PCA) in the gamma band distinguished ADHD from healthy subjects. Cluster coefficient and path length values in the sigma band were also significantly different between epochs, indicating different spindle-related brain activity in ADHD.

## 1. Introduction

Attention deficit hyperactivity disorder (ADHD) is a neurobehavioral disorder marked by a persistent pattern of inattention and/or hyperactivity-impulsivity that interferes with functioning or development. By definition, the symptoms are present in two or more life contexts, and there is clear evidence that they interfere with or reduce the quality of social, school, or work functioning [1]. Although the neurobiological pathogenesis of the disorder is longstanding and shared, no biological marker is available to date [2]. It has been proposed to integrate an EEG biomarker (theta/beta ratio) together with a standard clinical ADHD evaluation as support for diagnosis [3,4,5,6]. However, to date, the diagnosis remains essentially clinical, based on clinical assessment integrated with questionnaires and rating scales [2]. Several issues continue to affect the diagnostic process: the subjectivity of ADHD symptoms, the irregular correlation between classification scales and the overlap of attention and behavioral symptoms with other disorders. 

The study of connectomics, which focuses on analyzing the brain’s network structure, is shedding light on the underlying causes of neuropsychiatric disorders such as ADHD. As more and more research is conducted in this field, novel discoveries are leading to greater understanding and advancements. Recognizing the fundamental value of dynamical changes in network functioning, the lens of connectomics is attractively orienting its focus from the study of the resting state to the analysis of time variation of brain functioning. Specifically, the study of brain dynamics is trying to translate some key concepts, such as those of adaptability and flexibility, commonly used by clinicians to define mental health, into quantifiable measures able to characterize normal brain functioning and differentiate physiological from pathological conditions [7,8]. Concerning ADHD, the application of quantitative analysis of EEG, particularly by means of measures of functional connectivity, has recently reopened the debate on the usefulness of EEG for the identification of a reliable biomarker [9,10,11,12,13,14,15,16,17,18].

Sleep EEG connectivity is much less studied in this field, even if the association between sleep disorders and ADHD is well-known in clinical practice and currently reported in the literature [19]. 

Studies discussed in the literature suggest that the borders between cortical and sleep maturation are blurred in ADHD and that further investigations are needed to better understand the relationship between sleep and neuropsychiatric symptoms. 

In cognitive neuroscience, spindle activity, the physiological electroencephalographic hallmarks of non-REM (NREM) sleep microstructure, have been most studied as particularly involved in synaptic remodeling. Sleep spindles are defined as trains of distinct waves with frequencies of 11–16 Hz (most commonly 12–14 Hz) and a duration of >0.5 s [20]. Known to be involved in important sleep-dependent cognitive processes, such as memory consolidation, sleep spindles have been studied as an indicator of cognitive and emotional-behavioral development in children [21,22,23]. 

Alterations in spindling profile have been linked to different neuropsychiatric disorders, suggesting that spindle hyper- and hypofunction can both contribute to the generation of clinical symptoms and, in some cases, could also play a decisive role in the etiology of the disease [24,25,26,27,28,29]. Moreover, to date, there is a growing body of literature on the description of sleep-dependent mechanisms of brain plasticity and the identification of possible relationships with cortical maturation and neurodevelopmental disorders [30]. Nevertheless, knowledge about the role of specific aspects of sleep microstructures in ADHD is still lacking, as sleep in ADHD has been mainly studied in terms of quality and continuity.

Aiming to fill this gap, we conceived a preliminary study on sleep EEG sensor-space connectivity in children with ADHD, focusing on spindle activity. 

## 2. Materials and Methods

### 2.1. Subjects

The study included eight (six males, two females) children with ADHD and nine (six males, three females) healthy children ranging in age from 7 to 11 years (9 ± 1.67 years for ADHD, 8 ± 1.22 years for healthy children). All healthy and ADHD children were recruited among children admitted to the Neuropsychiatric Department of the Institute for Maternal and Child Health “Burlo Garofolo” (Trieste, Italy). The diagnosis of ADHD was made according to DSM-5 criteria [1]. None of the patients was pharmacologically treated at the time of the EEG examination. Healthy subjects, now referred to as controls, were admitted to the hospital for a suspected neuropsychiatric symptom later excluded by the negativity of the clinical examination and laboratory investigations; inclusion criteria for controls were normal psychomotor development, normal neurological examination, and normal EEG. The parent’s/legal guardian’s written informed consent was obtained for each participant.

### 2.2. EEG Recordings and Pre-Processing 

The EEG was recorded by means of nineteen surface electrodes placed according to the 10–20 International System. The signals were acquired using a computerized system (Micromed S.p.A., Mogliano Veneto, Italy) with the earlobe reference. EEG signals were acquired at a sampling frequency of 512 Hz, and during acquisition, the impedances were kept below 5 kΩ. 

Subsequently, EEG signals were digitally filtered with the second-order Butterworth band-pass filter within the 0.5–45 Hz frequency range. EEG was recorded in the early afternoon after sleep deprivation. The day before, the children were asked to go to sleep later than usual (after midnight) and wake up early (5:00–6:00 a.m.). Acquisition started at about 01:30 p.m. and lasted about 60 min. All children fell asleep during the EEG recording. Only the 2nd stage NREM sleep phase was included in the analysis. In particular, EEG recordings were visually inspected by two of the trained authors (M.C. and C.Z.) to identify the 2nd stage NREM sleep phase and the start and end points of each spindle free of artifact. To confirm the identification of spindle parameters, the inspection of EEG was performed twice and separately by each of the mentioned authors. Sleep phases were identified according to AASM criteria [20].

In order to make the analysis of spindle dynamics easier and more reliable, we developed by means of PSD a semi-automatic method able to identify the three epochs only requiring the visual identification of the spindle. In particular, for each spindle, epochs lasting 1 s were selected as follows: first, the maximum power peak in the sigma band (11–15 Hz) of the Fz channel was calculated and defined as the center of the during (E2) spindle epoch; the EEG was then cut 0.5 s before and after the maximum power peak in order to obtain the spindle epoch (E2). Next, the times corresponding to when the power reached 95% of the maximum power peak before and after the spindle were determined. Based on these times, one-second epochs were extracted to yield the epochs before (E1) and after (E3) the spindle.

In order to control the non-overlapping of the epochs between two consecutive spindles, the epochs have been visually inspected. The number of spindles in each subject varied from 6 to 10. For each subject, every single spindle was considered as a trial of the same stochastic process. In order to account for sample size bias and to obtain a more robust and reliable estimate, the bootstrap method [31,32] was then applied to each epoch to obtain 100 new sequences for each subject (Total of 100 × 18 subjects × 3 epochs). The steps of data processing are summarized in Figure 1. The first step was the aggregation of EEG data in matrices, then used as a variable for the evaluation of Multivariate autoregressive models (MVAR) [33]. The matrices are directly used to evaluate the MVAR model; since EEG signals could be nonstationary [34,35], the Adaptive MVAR (see Appendix A for more details) of SIFT Matlab toolbox was applied [36,37].

### 2.3. Generalized Partial Direct Coherence (gPDC)

The coefficients of the models were used to calculate the Generalized Partial Direct Coherence (gPDC) values. PDC between channel *i* and channel *j* is a frequency-domain approach that describes the directional flow of information from channel *j* to *i* at a frequency *f*, normalized by the total outflow information from *j* to the other channels [38]. Previous studies have confirmed its higher specificity and its tolerability to low signal-to-noise ratio (SNR) than other connectivity measures [39]; a significant strength of the PDC technique is that it gives information about the direction of the connections. The gPDC is a type of frequency domain representation of Granger causality via PDC that improves variance stabilization. The main advantage of gPDC is its hugely reduced variability for short time series [40]. We used the gPDC method to show the degree of directional linear interdependence between pairs of EEG channels during E1, E2 and E3 epochs. 

The gPDC has been performed on the full set of spindles separately for each epoch of a single subject, using an appropriate routine from SIFT Matlab toolbox [36,37] in order to represent the full set of simultaneously observed time series (Figure 2). The gPDC matrices were analyzed in six EEG frequency bands: delta (0.5–4 Hz), theta (4.5–8 Hz), alpha (8.5–13 Hz), beta (13.5–30 Hz), sigma (11–15 Hz) and gamma (30.5–45 Hz) bands. In each subject, we calculated the mean gPDC for each epoch (E1, E2 and E3). Based on gPDC, for each frequency band, in each epoch, we obtained a weighted connectivity matrix both for each original and bootstrapped sequence. The weighted matrices of bootstrapped sequences were used to evaluate the distribution of the connectivity values for each band in each group without distinction among epochs. Therefore, we obtained six distributions for each group, which were used to calculate the thresholds. The thresholds were chosen as the fiftieth percentile of their respective distributions (see Appendix A). 

Binary connectivity matrices were obtained by applying the thresholds to each gPDC matrix of epochs and bands of the original subjects (for more details see Appendix A). By means of the BCT Matlab toolbox [41], we used the binary graph matrices to calculate for each band and epoch the in (K*_i_*in) and out (K*_i_*out) node degrees, the cluster values of single channels (C*_i_*), the mean cluster (C) value and the shortest path length (L).

### 2.4. Statistics

In order to evaluate statistical differences between ADHD and controls, the Wilcoxon rank sum test for independent samples was applied to all the parameters (connection number, K_i_in, K_i_out, C, C_i_ and L); meanwhile, the comparison between pairs of epochs was evaluated by applying the Wilcoxon sign rank test for dependent samples on connection number, C and L parameters. Differences with a *p*-value < 0.05 were defined as significant, and in the case of the analysis among epochs, the Bonferroni correction for multiple comparisons was applied.

## 3. Results

### 3.1. Number of Connections

Considering the three epochs, E1, E2 and E3, the gamma band shows a greater number of connections (NC) compared to all other bands, mostly in the control group (Figure 3). The NC increases during the spindle epoch with a similar inter-epochs dynamic in the two groups for all the bands, except for the delta and gamma bands: in the delta band, the number reduces in both groups; in the gamma band, it reduces only in the control group. 

The NC in all three epochs was statistically different between the two groups in the delta and gamma band: it was higher in the delta band in the ADHD group and lower in the gamma band. In theta band, the difference was significant only in E2 (values in ADHD greater than in the controls). Since the main differences between groups were present in the delta and gamma bands, the graph analysis was focused on these two bands other than on the sigma band (the spindle band).

### 3.2. Nodes Degrees and Principal Component Analysis

The Principal Component Analysis (PCA) method was applied to K_i_in and K_i_out values obtained by averaging them among channels in the gamma band for each subject and epoch in order to evaluate the possibility of distinguishing the two groups by only using six variables. Finally, Fischer’s linear discriminant rule was used to find the line that divided the two groups based on the first two components.

The analysis of nodes’ degree and, in particular, the relation between “in-degree” K_i_in and “out-degree” K_i_out showed different behaviors during spindle in the three bands (Figure 4). The relation between K_i_in and K_i_out of each channel showed an inversion of the values between the two groups, passing from low frequency (delta) to high frequency (gamma) bands. In low frequencies, the values were lower in the controls than in ADHD, while in high frequencies, values in the controls were higher than in ADHD. In the sigma band, the values between the two groups were less segregated. The dynamic modifications across epochs showed a quite similar distribution of K_i_in and K_i_out within each group in delta and gamma bands.

However, in these bands, the distributions of K_i_in and K_i_out tend to cluster in quite separate “clouds”. This segregation is more evident in the gamma band.

In the sigma band, the number of K_i_in and K_i_out during the E2 epoch slightly increases, but the distribution of K_i_in and K_i_out shows less differentiation. Concerning differences in the K_i_in and K_i_out values of single channels, we did not identify specific topographic patterns in any band.

PCA was performed on the mean values of K_i_in and K_i_out. These means were obtained by averaging the values of all channels and epochs for each subject and for each band. The findings revealed that only in the gamma band the first two PCA components were able to clearly distinguish the two groups (Figure 5). 

### 3.3. Clustering Coefficient and Path Length 

Clustering coefficient (C) and Path Length (L) values showed different dynamics depending on the frequency band, even if with a similar inter-epochs trend in the two groups (Figure 6a). 

Significant differences for C and L values between the two groups in all three epochs were found in the delta and gamma bands, while in the sigma band, the differences, although present, were not significant. In particular, C values were higher in ADHD than in the controls in the three epochs of the delta and sigma bands, while in the gamma band, the highest values were observed in the controls. On the other hand, L values showed an opposite behavior in all the epochs and bands. Furthermore, C values showed significant differences among epochs only for ADHD between E1 and E2 in the delta band and between E2 and E3 in the gamma band. In the sigma band, both C and L values showed significant differences between E2 and the other two epochs for both ADHD and controls, with the highest C values and lowest L values in E2 in both groups. 

The results of the analysis carried out on single channels cluster C_i_ (Figure 6b) highlighted many significant differences between the two groups in all three epochs in the delta and gamma bands; in the sigma band, few differences were found only during E2. In the gamma band, cluster values of controls were greater than ADHD in almost all channels, while in delta and sigma bands, values were higher in ADHD.

## 4. Discussion

The main results of this study can be summarized as follows: (1) sleep EEG functional connectivity, analyzed as related to spindle activity, shows significant differences in children with ADHD; (2) differences in connectivity parameters depend on frequency bands; in particular, the gamma band mostly distinguish children with ADHD from children with typical neurodevelopment. These findings are discussed in detail in the following subchapters.

### 4.1. Neurophysiological Characteristics and Functional Connectivity—Sleep EEG in ADHD

Previous studies that analyzed connectivity in ADHD were carried out in a resting state or after a specific task [42,43,44,45], while there is a lack of studies on connectivity during sleep.

The results of our study show that connectivity parameters, such as the number of connections (NC), clustering coefficient (C) and path length (L), calculated in different epochs of N2 sleep, highlight dynamic modifications related to spindle activity and allow to identify the three phases (before, during and after the spindle activity) and to differentiate ADHD from controls.

In particular, in the sigma, theta, alpha and beta bands, the number of connections increases during the spindle (Figure 2) and, in parallel, the C value increases and the L value reduces. Moreover, only in the sigma band and partially in the alpha band (as was expected since the alpha band is included in the sigma band), all three parameters are significantly different during spindle activity (E2) with respect to activity before (E1) and after (E3) spindle activity in both populations, being present in the delta and gamma band only slightly changes during E2 (Figure 6a). 

In delta and gamma bands, the values of the three parameters are significantly different between the two groups in all three epochs. In particular, higher C values and lower L values for ADHD were found in the delta band and higher C values and lower L values for controls in the gamma band; the number of connections (NC) in controls showed lower values in the delta band and higher values in the gamma band. Furthermore, in the gamma band, the relation between K_i_in and K_i_out degrees (Figure 3) clearly shows that the mean values for single channels among the ADHD subjects are well distinguished from those of the controls in all three epochs. In the other two bands, these differences are less evident. Hence, the results of PCA suggest that the gamma band seems to be the frequency band in which the two groups could be distinguished by using either the NC, C and L parameters or K_i_in/K_i_out values. 

In in vivo studies, gamma oscillations are demonstrated to emerge from the coordinated interactions of excitatory and inhibitory neurons [46]. Gamma band activity is generally associated with cognitive processing and consciousness on awaking, and it is considered to reflect the processing of memory in local cortical networks. During sleep, cortical high-frequency activity has been alleged to have an important role in the generation of spindles [47]. Gamma frequency-salient abnormalities between children with ADHD and typically developing children have been recently reported in a study on NREM stage 1 EEG functional connectivity [48]. It has been suggested that neural connectivity disturbances may arise from alterations in the balance between glutamatergic excitatory and GABA-ergic inhibitory signaling at the microcircuit level [18]. These alterations could possibly reflect a lower effectiveness of the brain networks in spindle generation and spindle role in cognitive functioning. 

### 4.2. Dynamic Modifications of the Neural Network in ADHD and Controls across Frequency Bands: Different Balances between Different Functions?

We found in the sigma band similar spindle-related connectivity dynamics in the two groups, with an increase of NC during spindle and opposite trends of C and L values (higher C values and lower L values in spindle epoch with respect to the other two epochs, in a sort of low-high-low C trend and high-low-high L trend). It seems that, in both groups, when the neuronal system generates the spindle, the network better segregates (higher clustering coefficient) and integrates (shorter path length) information approaching a so-called small-world functioning [49], and this seems more evident in the control group. 

In the last years, in the field of connectomics, the concept of small-worldness has gained growing interest; as one of the most influential findings about brain functioning, it is considered that human brain networks exhibit prominent small-world organization. Such a network architecture would facilitate efficient information segregation and integration at low wiring and energy costs, presumably resulting from the natural selection under the pressure of a cost-efficiency balance, undergoing continuous changes during normal development and exhibiting significant alterations in neurological and psychiatric disorders [50].

Recently, this concept has been used to describe network organization of sleep EEG activity: a trend has been reported on the functional connectivity during sleep to move forward to an organization more similar to that of a small-world network, with different small-world properties and different trends in the EEG frequency bands [51].

In ADHD, sleep difficulties have been mostly viewed as an epiphenomenon of an underlying maturational disorder [52,53,54,55,56]. The role of sleep in ADHD has been mainly studied in terms of quality and continuity of sleep [19,54,55,56,57,58]. 

Recently, the presence of small-world variations across different frequency bands has been described with fMRI in resting state [59,60,61], findings still lacking from sleep EEG.

In our study, we found that compared to the sigma band, connectivity in the delta band seems less modified by the spindle activity, maintaining quite constant values of C and L, with a functioning approaching that of a small world in the control group. The gamma band show opposite dynamic changes in the two groups: in the control group, the network seems to move toward a small-world functioning (the values of C are the highest compared to the other two bands, and the values of L are the lowest), while in ADHDs there seems to be a weaker small-worldness (since the values of C are similar to those of delta, but the values of L are slightly lower). 

Globally considered, our findings seem to reveal simultaneous different dynamical functioning of the neural network to the arrival of the spindle, depending on the frequency band. This could represent another aspect of the continuous modulation across different brain states of cortical connectivity during sleep, as hypothesized by Vecchio et al. [51]. Dynamic changes in functional connectivity across sleep phases are an expanding field of research, and complex modulation of connectivity patterns with the deepening of NREM sleep has been recently described [62].

According to Steriade et al., slow oscillations during sleep have the ability to trigger and group faster cortical oscillations such as spindles and high-frequency activities in the beta or gamma range [63]. It has been hypothesized that gamma oscillations might be associated with phasic increases in neural activity during slow oscillations and could play a role in the offline processing of cortical networks [64]. Concerning low-frequency activity, Cox et al. recently provided a comprehensive view of how human slow oscillatory dynamics influence various measures of brain processing [65]. 

Concerning C differences in the topography of single electrodes, we found that in ADHD, in the delta band, especially during the spindle epoch, C values of almost all the electrodes were higher than in the control group, while in the gamma band, C values were higher in the control group, in all the three epochs. No specific topographic pattern was found in any band. 

Spindles are hypothesized as originating from the thalamic reticular nucleus and rhythmic oscillations within the thalamocortical circuits, but little is known about their spatial properties and how they could influence functional connectivity of cortical and subcortical activities and modify brain networks organization and functioning. Recent works suggest that spindles are not homogenous, leaving unclear to what degree spindles are global or local and if their properties are uniform or location-dependent [66]. It has been proposed that the combination of local characteristics and global organization reflects the dual properties of the thalamocortical generators and provides a flexible framework to support the many functions ascribed to sleep in general and spindles specifically [66].

Mills et al. [67] emphasized the role of the thalamus as a mediating structure and suggested that when functional connectivity is altered, this is likely related to altered communication between the thalamus and subcortical areas, as well as altered communication between the thalamus and the cortex, thereby resulting in dysfunctional regulation of behavior.

Our results, showing different spindle-related sensor-space connectivity dynamics in the two populations, in the delta but especially in the gamma band, raise questions about the involvement of the thalamus in ADHD and the role of gamma activity during sleep suggesting the presence in children with ADHD of an altered communication between the thalamus and the cortex, maybe mediated by an imbalance in the way delta and gamma activity interact.

As a preliminary study, we are aware of its limitations and careful in overgeneralizing results. Mainly the low number of subjects and the non-consistent EEG connectivity methodological framework across the literature [68,69] make us cautious in drawing conclusions. Even though we tried to follow the recommendation and apply the surrogate data methods based on phase randomization and bootstrap procedures [31,32,33,70], with the purpose of automatic connectivity thresholds selection, the metrics for information loss used in the procedure are not yet fully standardized [68]. In addition, due to the nature of the experimental setup, the data were acquired on a clinically standard nineteen channels EEG system. The analysis was performed on EEG sensor space due to the limited number of channels and thus can be affected by the volume conduction as a result of a linear mixture of source activities from all brain regions. Even though the study on cortical source analysis of resting state EEG found that both approaches resulted in concordant findings presenting similar trends of changes in resting EEG [71], it can still lead to spurious correlations between the signals of short-distance electrodes in sensor space connectivity analysis.

However, to the best of our knowledge, there are no previous studies on sleep spindle connectivity in children, and the results of this study encourage us to further investigate this field, looking for a possible biomarker of pathological brain functioning in ADHD.

In conclusion, connectivity methodologies focused on EEG microstructural elements have not yet been widely addressed and could possibly be able to provide valuable insight into this field.

The present preliminary study’s results encourage further investigation of sleep EEG functional connectivity, particularly its dynamics related to spindle activity. If confirmed in larger clinical studies, these results could support the usefulness of an easily achievable, not invasive, and low expensive method to helping clinical diagnosis of ADHD. The study suggests the coexistence of different spindle-related functional connectivity features depending on frequency bands, raising questions about the involvement of the thalamus in ADHD and the role of gamma activity during sleep. Traditionally considered a hallmark of the normal organization of brain functioning and development, sleep spindles could be conceptualized and analyzed as a new window to study brain networks’ adaptability to internal as well as external events.

## Figures and Tables

**Figure 1 entropy-25-01244-f001:**
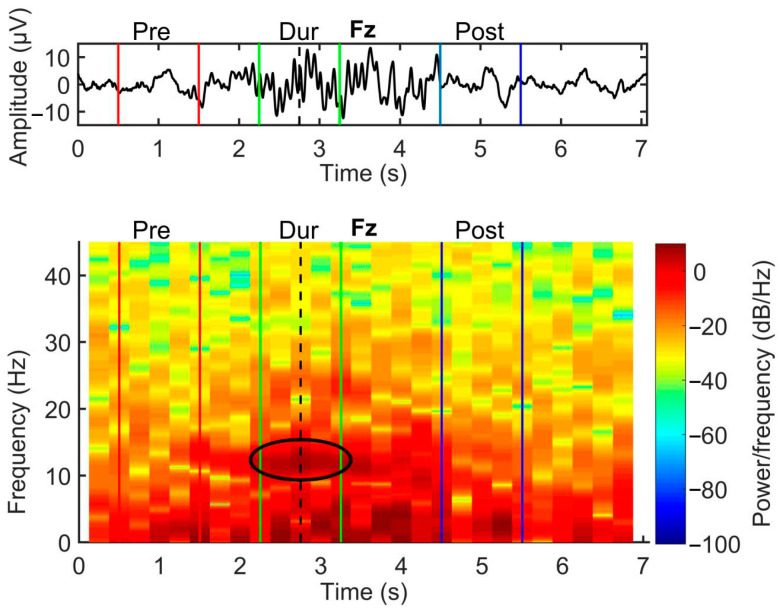
The top panel represents the EEG of a spindle in the Fz channel. Red vertical lines identify the interval of the E1 pre-spindle epoch, green vertical lines the E2 spindle epoch and blue lines the E3 post-spindle epoch. The bottom panel shows the power of the same EEG signal with colors corresponding to power amplitude. The black ellipse highlights the frequency band (11–15 Hz) related to the spindle, in which a peak of power value is present.

**Figure 2 entropy-25-01244-f002:**
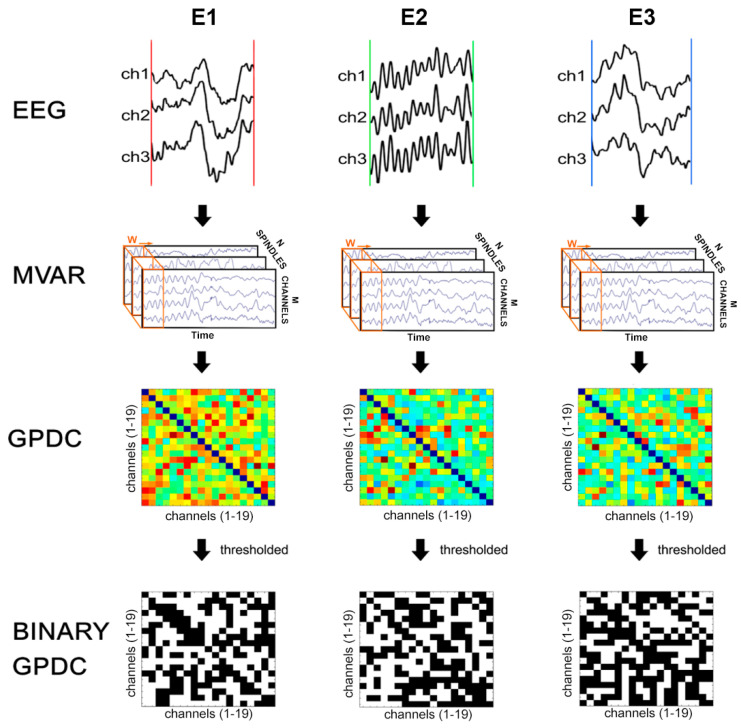
Steps of sleep EEG epochs data processing (MVAR = multivariate autoregression, GPDC = generalized Partial Direct Coherence).

**Figure 3 entropy-25-01244-f003:**
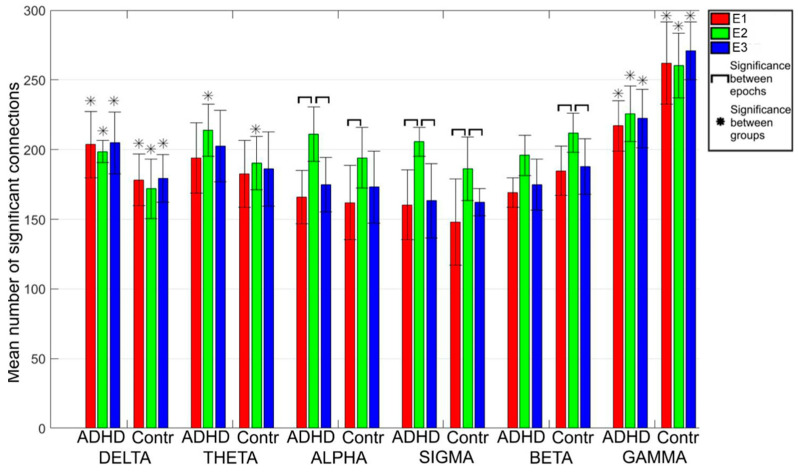
Bars indicate the mean number of connections for each band in the three epochs: E1 (pre-spindle), E2 (spindle) and E3 (post-spindle), with the standard deviations, in children with ADHD and in Controls. The symbol * highlights significant statistical differences between the two groups; horizontal lines indicate differences among epochs within each group.

**Figure 4 entropy-25-01244-f004:**
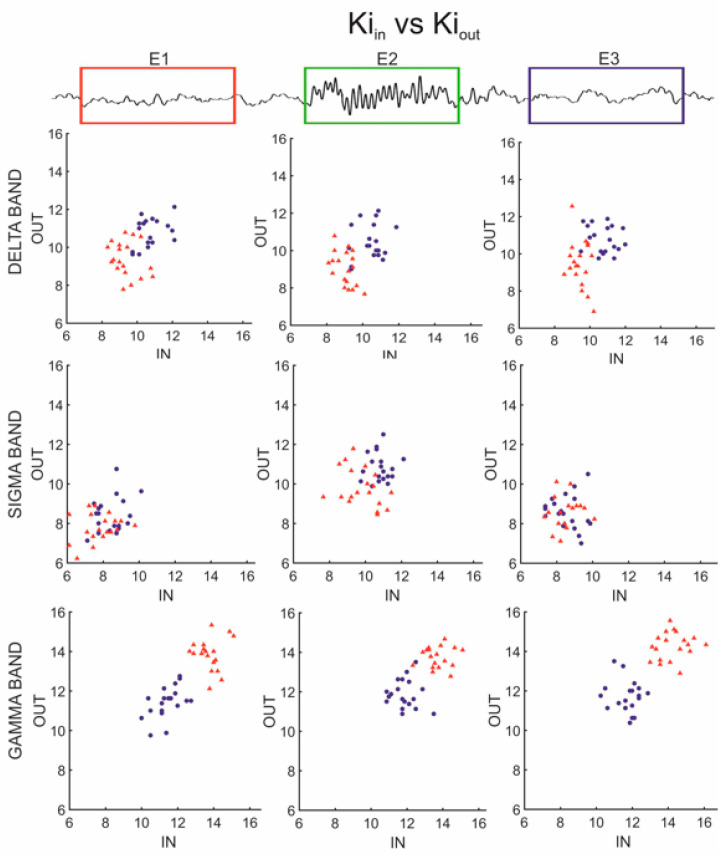
Representations of “in-degree” (K_i_in) values plotted versus K_i_out in low (delta), middle (sigma) and high (gamma) frequency bands for the three epochs: on the left, the pre-spindle epoch (E1), at the center the spindle epoch (E2), on the right the post-spindle epoch (E3). Red triangles represent channels in the Controls group; blue circles represent channels in the ADHD group.

**Figure 5 entropy-25-01244-f005:**
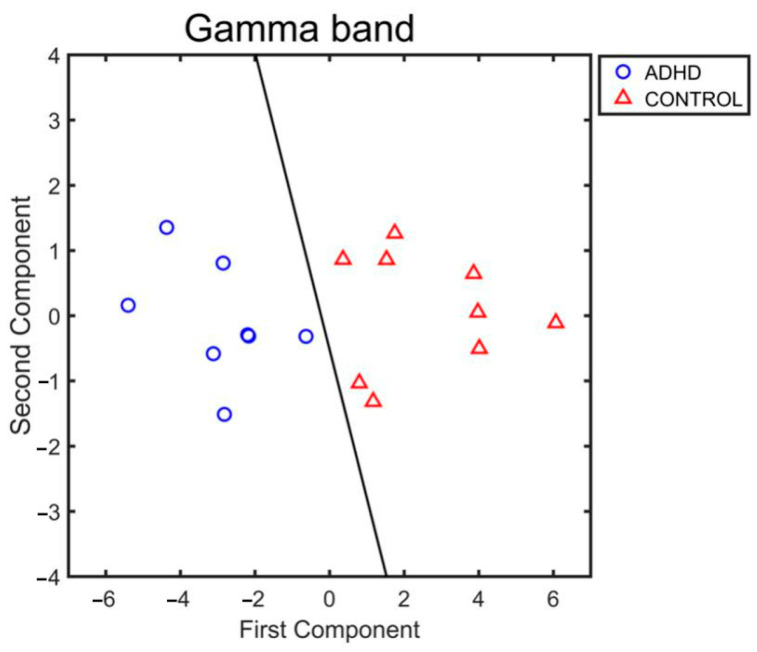
Representation of the first two components resulted from Principal Component Analysis (PCA) in the gamma band. Red triangles represent Controls group, blue circles ADHD group. Black line is the result of the application of the Fisher linear discriminant rule evaluated as *K + L*(1)*x*_1_
*+ L*(2)*x*_2_
*=* 0.

**Figure 6 entropy-25-01244-f006:**
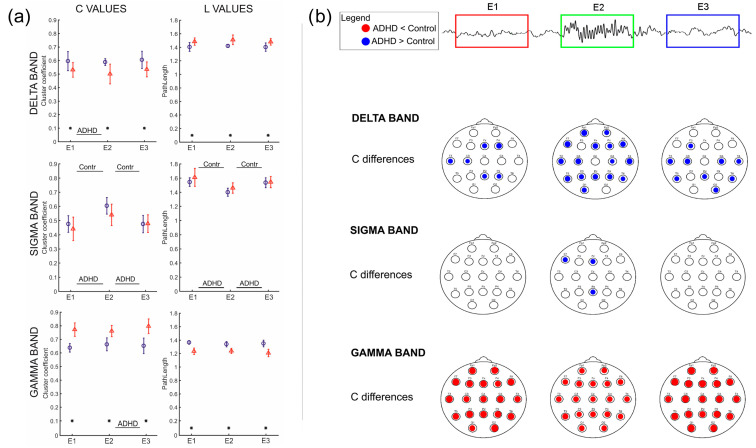
(**a**) Mean values of C and L parameters, for each epoch, in delta, sigma and gamma bands. Red triangles represent the Control group, and blue circles the ADHD group. Vertical bars on the markers show the standard deviation of the values. The symbol * represents the significant difference between the two groups; black lines above (Controls significances) or below (ADHD significances) the markers show statistical significances between epochs in the two groups (**b**). Significant differences of single node clustering coefficient (C*_i_*) between the two groups in all three epochs (pre-spindle E1, spindle E2 and post-spindle E3) in Delta, Sigma and Gamma bands. Fulfilled circles represent the channels with *p* < 0.05. Blue circles mean ADHD values greater than Controls, and red color vice versa.

## Data Availability

The data presented in this study are available on request from the corresponding author. The data are not publicly available due to privacy restrictions.

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
