# Peer review of "Sleep Spindle-Related EEG Connectivity in Children with Attention-Deficit/Hyperactivity Disorder: An Exploratory Study"

_entropy, 2023, doi:10.3390/e25091244_

Round 1
Reviewer 1 Report
This work explored the role of sleep in ADHD by using an network architecture to facilitate efficient information segregation and integration. The concept was used to describe the organization of sleep EEG activity, with trends of the functional connectivity during sleep moving towards a small-world network with different small-world properties and different trends in the EEG frequency bands. The data was processed by aggregating the EEG data into matrices and using Multivariate autoregressive models (MVAR) to evaluate the matrices. And selection between control and ADHD group based on parameter which are defined as features.
The paper is well written and acceptable for publication in the MDPI Entropy journal. The following are suggested comments to improve the quality of the paper:
· Based on EEG signals, it is possible to determine the state of sleep or anesthesia. It would be beneficial to identify common sleep patterns among patients and collect data from comparable stages of sleep across all patients. collecting data only based on time passed are not recommended.
· The author failed to localize the source during measurement before initiating functional connectivity. While some similar research has been conducted without source localization, using the term "functional connectivity" without source localization usually refers to surface potential feature selection and is not recommended.
· It would be beneficial to include more recent papers as references, as most of them were published before 2017.
· Please include the details of your method and its relevant mathematical concepts in the Appendix. It would be great if you could also share a Git repository containing your code and sample data.
Author Response
Reviewer 1
This work explored the role of sleep in ADHD by using an network architecture to facilitate efficient information segregation and integration. The concept was used to describe the organization of sleep EEG activity, with trends of the functional connectivity during sleep moving towards a small-world network with different small-world properties and different trends in the EEG frequency bands. The data was processed by aggregating the EEG data into matrices and using Multivariate autoregressive models (MVAR) to evaluate the matrices. And selection between control and ADHD group based on parameter which are defined as features.
The paper is well written and acceptable for publication in the MDPI Entropy journal. The following are suggested comments to improve the quality of the paper:
>> Dear Reviewer,
Thank you for taking the time to review our work. We appreciate your valuable feedback and suggestions to enhance the quality of our paper. Your insights have been immensely helpful in further improving our research.
- Based on EEG signals, it is possible to determine the state of sleep or anesthesia. It would be beneficial to identify common sleep patterns among patients and collect data from comparable stages of sleep across all patients. collecting data only based on time passed are not recommended.
>> Thank you for the comment. The EEG acquisitions were performed in the pre-established time of day interval, but only the 2nd stage NREM sleep phase was included in the analysis. We modified the manuscript to be more specific regarding this point.
“The day before, the children were asked to go to sleep later than usual (after midnight) and wake up early (5.00-6.00 am). Acquisition started at about 01.30 p.m. and lasted about 60 minutes. All children fell asleep during the EEG recording. Only the 2nd stage NREM sleep phase was included in the analysis. In particular, EEG recordings were visually inspected by two of the trained authors (M.C. and C.Z.) to identify the 2nd stage NREM sleep phase and the start and end points of each spindle free of artefact. To confirm the identification of spindle parameters, the inspection of EEG was performed twice and separately by each of the mentioned authors. Sleep phases were identified according to AASM criteria [14]. “
- The author failed to localize the source during measurement before initiating functional connectivity. While some similar research has been conducted without source localization, using the term "functional connectivity" without source localization usually refers to surface potential feature selection and is not recommended.
>> Thank you for the comment. The source-based analysis of functional connectivity would add additional value to this study. However, due to the limited number of channels and potential susceptibility to noise in this particular experimental setting, we conducted a sensor-space analysis of functional connectivity. We specified better in the manuscript (in the abstract and introduction) that we performed sensor space connectivity analysis. In addition, we added a comment regarding this limitation.
- It would be beneficial to include more recent papers as references, as most of them were published before 2017.
>> Thank you for the comment. We updated the literature review with more recent studies.
- Please include the details of your method and its relevant mathematical concepts in the Appendix. It would be great if you could also share a Git repository containing your code and sample data.
>> Thank you for the comment. We added the supplementary material with relevant mathematical concepts, as suggested. We are considering to make available the scripts and sample data in the future.

Author Response
We would like to thank you for your time to review our work. Your feedback and suggestions have been valuable in helping us enhance the quality of our paper. Please find our responses in the attachment.

Round 2
Reviewer 2 Report
While I don’t fully agree to the application of Wilcoxon testing in favor of network sparsity, I can see the merit in their response.
Apart from that, the authors answered sufficiently all my comments.
My only comment is the title. “Do hyperactive children wish of connected spindles” is not relevant to anything written in the manuscript and vague (either not clearly expressed or understood). I would suggest to remove it or modify it to something like “ a preliminary study in hyperactive children”.
Author Response
Dear reviewer,
Thank you for your comment. Following your recommendation, we modified the title to:
"Sleep spindle-related EEG connectivity in children with Attention-Deficit/Hyperactivity Disorder: an exploratory study"